# Exploring antecedents and outcomes of burnout among emergency department staff using the job demands-resources model: A scoping review protocol

**Luhuan Yang[1], Yunhong Lei[2], Dongmei Chu[1], Jiawei Jiang[1], Zifeng Li[3], Yanhua Tang[1], Abd Rahman Anita** [1] *

**1** Faculty of Medicine and Health Sciences, Department of Community Health, Universiti Putra Malaysia, Serdang, Selangor, Malaysia, **2** Philippine Women's University School of Nursing, Manila, Philippines, **3** Department of Traditional Chinese Medicine, The First College of Clinical Medical Science, Three Gorges University/Yichang Central People's Hospital, Yichang City, Hubei Province, China

\* anitaar@upm.edu.my

**Data Availability Statement:** No datasets were generated or analysed during the current study.

## Abstract

### Introduction

High levels of burnout are prevalent among Emergency Department staff due to chronic exposure to job stress. There is a lack of knowledge about anteceding factors and outcomes of burnout in this population.

### Aims

To provide a comprehensive overview of burnout and identify its workplace antecedents and outcomes among Emergency Department staff.

### Methods

The scoping study will follow the methodology outlined by the Joanna Briggs Institute. PubMed, Scopus, Web of Science, APA PsycInfo, and CINAHL databases will be searched using predefined strategies. Two reviewers will screen the title, abstract and full text separately based on the eligibility criteria. Data will be charted, coded, and narratively synthesized based on the job demands–resources model.

### Conclusion

The results will provide insights into the underlying work-related factors contributing to burnout and its implications for individuals, healthcare organizations, and patient care.

Deidentified research data will be made publicly available when the study is completed and published.

**Funding:** The author(s) received no specific funding for this work.

## Introduction

Globally, healthcare workers in emergency departments (EDs) are experiencing increasing physical and emotional challenges [1,2]. Factors contributing to this stress include complex patient loads, long shifts, a broad range of diseases, insufficient human resources, fast-paced work, unpredictable working conditions, and heavy rescuing and managing tasks [3–5]. These working conditions, known as job stressors, significantly impact the staff. In turn, the individual responses of workers to these stressors, termed job strain, are of critical concern. Many studies indicate that cumulative strain can lead to "burnout" characterized by emotional exhaustion, depersonalization, and reduced professional efficacy [6–8]. Currently, ED healthcare workers are struggling to cope with an ever-escalating burnout syndrome, with recent studies showing alarmingly high prevalence rates [9]. While burnout rates in residents of different specialties lie between 18% and 80%, the rates among emergency healthcare workers range from 32% to 60% [10]. A systematic review of 17 studies revealed that, on average, approximately 26% of emergency nurses suffer from burnout [11]. Additionally, research focusing on ED physicians in Canada has shown that 46% experience medium-to-high levels of burnout [12]. Furthermore, a sample of Spanish healthcare workers in the ED exhibited burnout prevalence rates of 57% [13]. Moreover, the emergence of the COVID-19 pandemic has further exacerbated the issue, making the syndrome of burnout even more prevalent and visible within these healthcare settings [14].

Burnout can lead to a variety of negative outcomes, such as health problems, conflict within the workplace, poor coping skills, and substance dependence [15,16]. In healthcare environments, when employees experience burnout, it often leads to higher turnover rates, increased absenteeism, poor job performance, and a general decline in morale[17]. Additionally, when healthcare workers experience burnout, it tends to affect the level of care they provide, leading to a decreased sense of patient safety and an increased likelihood of medical errors [18–20]. The potential adverse outcomes of burnout have generated widespread interest and emphasized the importance of understanding burnout in ED healthcare workers.

Existing models for occupational stress offer valuable frameworks for elaborating burnout by identifying common workplace antecedents and outcomes. Among these, the Job Demands-Resources (JD-R) model, proposed by Bakker and Demerouti, stands out for its comprehensive approach [21]. This model is widely recognized as the most prevalent job stress model, effectively explaining occupational stress and its impact on employee health and organizational outcomes. The JD-R model assumes that despite the diversity of work environments across various organizations, each occupation generates distinct job stressors that can be categorized as job demands and resources.

Job demands encompass various physical, psychological, social, and organizational dimensions, requiring sustained psychological and physical effort or skills [22]. Increased levels of demand are predictors of burnout and are linked to low engagement and increased turnover [23]. Job resources are those aspects of a job that facilitate the individual's ability to achieve job goals, promote personal growth and reduce job demands [22,24]. The fundamental premise of the JD-R model is that, regardless of the specific job or occupation, job stress appears in the context of increased demands of particular jobs and constraints on specific job resources, which in turn impacts psychological well-being and organizational outcomes.

To the best of our knowledge, several reviews in the past decade have described the prevalence and determinants of burnout among ED staff, but have not investigated individual and organizational outcomes in detail. A systematic review by Jef Adriaenssens et al. assessed the prevalence of burnout in emergency nurses and identified specific (individual and work-related) determinants of burnout in this population [25]. However, only quantitative studies

published between 1989 and 2014 were included. Two systematic reviews [20,26] summarized studies on burnout among ED physicians, focusing mainly on the burnout rate. There is a lack of knowledge about anteceding factors in this population. This research aims to provide a comprehensive overview of burnout and its antecedents and outcomes among ED staff. It will shed light on these professionals' burnout challenges and underscore the urgency of addressing this issue. The study can also offer healthcare organizations vital information for crafting targeted interventions and fostering supportive work environments, thus improving healthcare workers' work conditions and quality of life in the EDs.

### Review objectives

This paper will aim to address three research objectives. Firstly, it seeks to systematically delineate past research endeavors in the domain, offering a comprehensive overview of burnout among healthcare personnel in the ED. Secondly, it aims to deepen our comprehension of the pre-existing conceptual framework that delves into potential antecedents and outcomes of burnout among ED healthcare workers. Lastly, the review strives to identify novel research gaps within the evidentiary foundation, mainly focusing on the scarcely investigated aspects of burnout among ED staff, encompassing both work-related antecedents and outcomes.

## Materials and methods

### Design

The scoping review will use the Joanna Briggs Institute (JBI) scoping review methodological framework [27,28]. This protocol followed the Preferred Reporting Items for Systematic Reviews and Meta-Analyses extension for Scoping Reviews (PRISMA-ScR checklist) [29] (S1 Appendix) and the Preferred reporting items for systematic review and meta-analysis protocols (PRISMA-P checklist) [30] (S2 Appendix). Additionally, the protocol was duly recorded and registered on the Open Science Framework (https://osf.io/ vfm63).

### Review questions

(a)What is the current research landscape related to burnout among healthcare personnel in the ED?

(b)How do different studies conceptualize and define the relationships between burnout, its work-related precursors, and subsequent outcomes within this specific occupational setting?

(c)Which antecedents and outcomes of burnout have received limited attention in the existing literature, necessitating further investigation?

### Identification of relevant studies

The PCC (Population/Concept/Context) framework was used to define retrieval strategy-related items. PubMed, Scopus, Web of Science, APA PsycInfo, and CINAHL databases will be searched. The search tactic combines Medical Subject Headings (MeSH) terms with keywords for healthcare workers, burnout, and emergency department (see **Table 1** for the Pubmed search strategy).

This study will cover all primary quantitative and qualitative mixed-methods studies that meet the eligibility criteria for the scoping review. Inclusion criteria include original, peer-reviewed articles in English, available in full-text, utilizing both quantitative and qualitative methods, that focus on workplace antecedents and outcomes of burnout among ED staff, from inception to August 19, 2023. Book chapters, conference abstracts, editorials, commentaries,

**Table 1. Full search string for the database "Pubmed".**

| | | Pubmed Search Strategy |
|---|---|---|
| **1. Population** | Subject Headings | "Health Personnel"[Mesh] |
| | | ("Health Care Provider*"[Title/Abstract] OR "Healthcare Provider*"[Title/Abstract] OR "Healthcare Worker*"[Title/Abstract] OR "Health Care Professional*"[Title/Abstract] ORHCWs[Title/Abstract] OR HCPs[Title/Abstract] OR "medical profession"[Title/Abstract] OR "medical worker*"[Title/Abstract] OR "medical staff" "health profession*"[Title/Abstract] OR "healthcare personnel"[Title/Abstract] OR "medical personnel"[Title/Abstract] OR "clinical staff"[Title/Abstract] OR paramedic*[Title/Abstract] OR doctor*[Title/Abstract] OR physician*[Title/Abstract] OR surgeon*[Title/Abstract] OR nurs*[Title/Abstract] OR technician*[Title/Abstract]) OR clinician[Title/Abstract])OR ((provider*[Title/Abstract] OR profession*[Title/Abstract] OR staff[Title/Abstract] OR workforce[Title/Abstract] OR practitioners[Title/Abstract] OR personnel[Title/Abstract] OR worker*[Title/Abstract]) AND (health[Title/Abstract])) |
| **2. Concept** | | *AND* |
| | Subject Headings | "Burnout, Psychological"[Mesh] |
| | Keywords | Burnout[Title/Abstract] OR Burn-out[Title/Abstract] or "Burn out"[Title/Abstract] |
| | | *AND* |
| **3. Context** | Subject Headings | "Emergency Service, Hospital"[Mesh] |
| | Keywords | "Emergency"[Title/Abstract] or "ER"[Title/Abstract] or "ED"[Title/Abstract] |

reviews, grey literature, and dissertations/theses will be excluded. Additionally, studies focusing on pediatric and pre-hospital emergencies will also be excluded.

## Data selection

Upon retrieval, all identified citations will be compiled and uploaded to Covidence (https://www.covidence.org/), a robust software solution for managing data screening and extraction in systematic reviews. Using the Covidence platform, two reviewers will independently carry out study selection, encompassing title and abstract screening, as well as full-text screening. Any uncertainty or discrepancies between the reviewers will be addressed through team deliberations. This process will follow the PRISMA diagram (See Fig 1).

## Data extraction

The data extraction template will record information from the included papers. The template will consist of items related to the author, publication year, journal, study objectives, setting, study design, samples, assessment of burnout among ED staff, and factors and outcomes. The framework will be adapted and refined as necessary. One reviewer will accomplish data extraction independently for all studies. Two more reviewers will then independently review the extracted data. In case of discrepancies or inconsistencies in interpretation between reviewers, a third reviewer will resolve these issues.

## Reporting the findings

The data will be descriptively analyzed using the completed extraction tool to map the available evidence. The key information will be systematically sorted and categorized to comprehensively summarize the evidence on burnout among healthcare workers in the ED. The acquired data will be visually represented through figures, charts, and tables. The report will describe

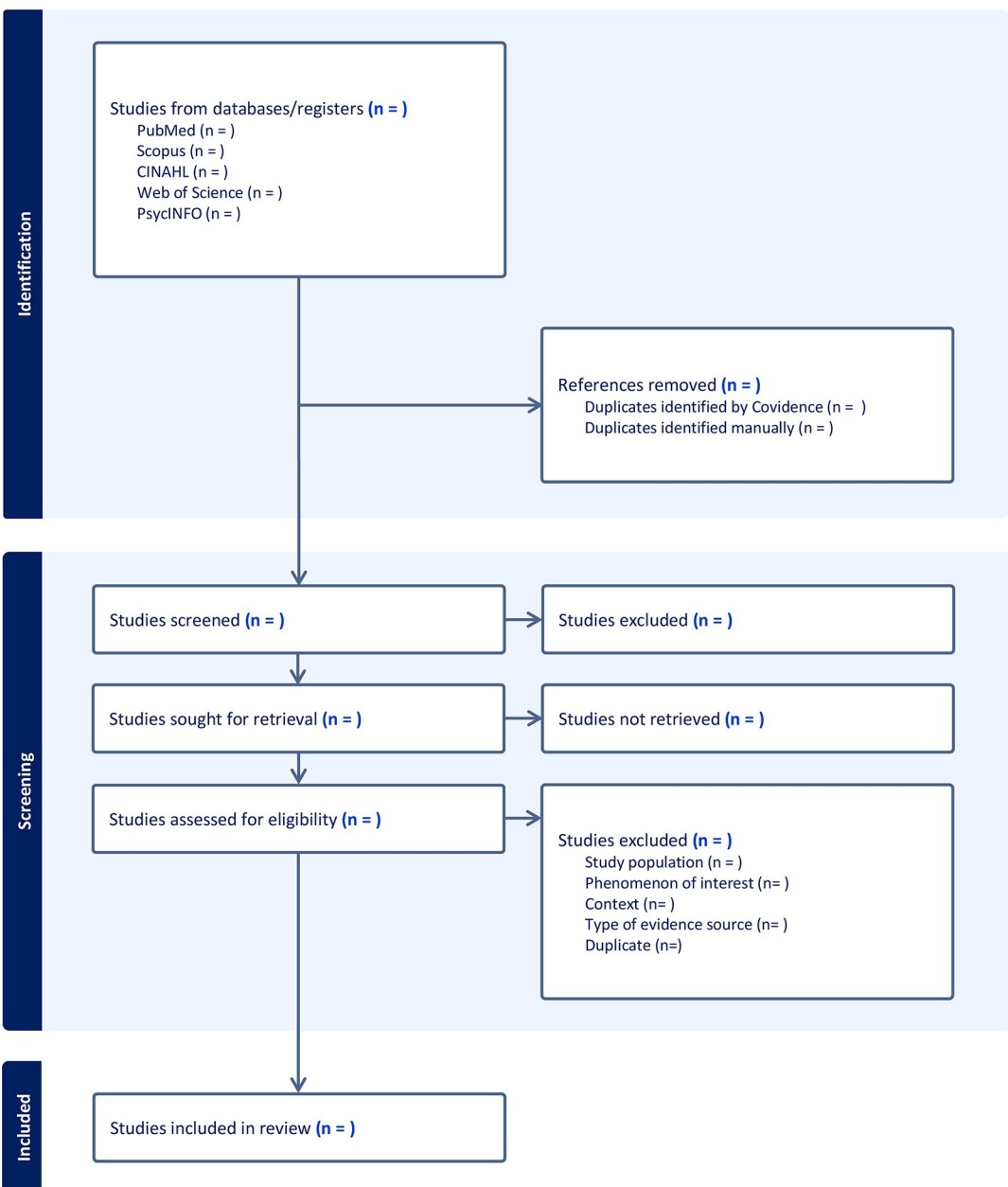

**Fig 1. Flow chart of study selection.**

the results concerning the research questions of the scoping review. Additionally, gaps in the literature will be identified, and potential implications for future research will be highlighted.

## Discussion and conclusion

The high prevalence of burnout among ED staff is a significant concern driven by chronic exposure to job stress in their demanding work environment. This scoping review protocol outlines the methodology for exploring the workplace antecedents and outcomes of burnout among ED staff using the JD-R model. By analyzing how different studies define and conceptualize these relationships, this study represents a significant step towards identifying patterns,

trends, and gaps in the existing literature, paving the way for a more holistic understanding of burnout within this population.

The implications of this review extend far beyond academic discourse. Healthcare professionals and researchers will gain a deeper insight into the nuances of burnout and its impacts. The identified gaps in antecedents and outcomes of burnout underscore the urgent need for strategic changes among healthcare policymakers and administrators. Targeted interventions are urgently needed to reduce specific job demands while enhancing job resources to reduce burnout among ED staff and create a healthier, more resilient, and more effective ED workforce.

## Supporting information

**S1 Appendix. Preferred Reporting Items for Systematic reviews and Meta-Analyses extension for Scoping Reviews (PRISMA-ScR) checklist.**
(PDF)

**S2 Appendix. Recommended items to address in a systematic review protocol*.**
(PDF)

## Author Contributions

**Conceptualization:** Luhuan Yang, Yunhong Lei, Abd Rahman Anita.

**Data curation:** Yunhong Lei.

**Investigation:** Luhuan Yang, Dongmei Chu, Jiawei Jiang, Yanhua Tang.

**Methodology:** Luhuan Yang, Dongmei Chu, Jiawei Jiang, Yanhua Tang.

**Software:** Luhuan Yang, Dongmei Chu, Jiawei Jiang, Yanhua Tang.

**Supervision:** Zifeng Li, Abd Rahman Anita.

**Writing – original draft:** Luhuan Yang.

**Writing – review & editing:** Yunhong Lei, Zifeng Li, Abd Rahman Anita.

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
