## [Decision Letter · Decision Letter 0]

14 Nov 2023

PONE-D-23-25522Exploring Antecedents and Outcomes of Burnout among Emergency Department Staff Using the Job Demands-Resources Model: Scoping Review ProtocolPLOS ONE

Dear Dr. Anita,

Thank you for submitting your manuscript to PLOS ONE. After careful consideration, we feel that it has merit but does not fully meet PLOS ONE’s publication criteria as it currently stands. Therefore, we invite you to submit a revised version of the manuscript that addresses the points raised during the review process.

We look forward to receiving your revised manuscript.

Kind regards,

Jasna Karacic Zanetti

Academic Editor

PLOS ONE

Journal Requirements:

- https://doi.org/10.1111/1742-6723.12135

In your revision ensure you cite all your sources (including your own works), and quote or rephrase any duplicated text outside the methods section. Further consideration is dependent on these concerns being addressed.

Additional Editor Comments:

Dear Authors,

We appreciate the manuscript you submitted. After a thorough evaluation by our esteemed reviewers, we are delighted to inform you that your manuscript requires only minor revisions.

The reviewers found your work to be promising, and with the suggested revisions, we believe it will significantly enhance the overall quality and impact of the publication.

Information regarding burnout could be found in articile Who Cares What the Doctor Feels: The Responsibility of Health Politics for Burnout in the Pandemic

link

https://www.mdpi.com/2227-9032/9/11/1550

Kindly review and evaluate the requested work to determine whether they are relevant and should be cited.

Once you have completed the revisions, please submit the revised version of your manuscript through our online submission system. Include both a clean version and a tracked changes version to aid our review process.

We eagerly anticipate the enhanced version of your manuscript and your continued collaboration with our journal.

Should you have any questions or require any clarification during the revision process, please do not hesitate to contact us.

Sincerely,

REVIEWER COMMENT TO BE ADRESSED:

The manuscript appears to be well structured, overall. A suggested minor edit is to offer a concise overview of the anticipated benefits and implications of the scoping review in the Introduction section. Explain how the review's outcomes could affect healthcare organizations or practitioners. This background can assist readers grasp the significance of your findings.

Reviewers' comments:

Reviewer's Responses to Questions

**Comments to the Author**

1. Does the manuscript provide a valid rationale for the proposed study, with clearly identified and justified research questions?

Reviewer #1: Yes

Reviewer #2: Yes

2. Is the protocol technically sound and planned in a manner that will lead to a meaningful outcome and allow testing the stated hypotheses?

Reviewer #1: Yes

Reviewer #2: Yes

3. Is the methodology feasible and described in sufficient detail to allow the work to be replicable?

Reviewer #1: Yes

Reviewer #2: Yes

4. Have the authors described where all data underlying the findings will be made available when the study is complete?

Reviewer #1: Yes

Reviewer #2: Yes

5. Is the manuscript presented in an intelligible fashion and written in standard English?

Reviewer #1: Yes

Reviewer #2: Yes

6. Review Comments to the Author

You may also provide optional suggestions and comments to authors that they might find helpful in planning their study.

Reviewer #1: The manuscript appears to be well structured, overall. A suggested minor edit is to offer a concise overview of the anticipated benefits and implications of the scoping review in the Introduction section. Explain how the review's outcomes could affect healthcare organizations or practitioners. This background can assist readers grasp the significance of your findings.

Reviewer #2: Introduction and Rationale: The introduction clearly states the high prevalence of burnout among ED staff due to chronic job stress. However, it would benefit from more specific data or statistics to emphasize the magnitude of the problem. Also, it would be useful to briefly mention why the Job Demands-Resources (JD-R) model is chosen for this study.

Methodology: The methodology section is thorough, explaining the use of the Joanna Briggs Institute methodology and the PRISMA-ScR checklist. However, it would be beneficial to include more detailed information on the criteria for selecting studies (e.g., year range, type of studies included).

7. PLOS authors have the option to publish the peer review history of their article (what does this mean?). If published, this will include your full peer review and any attached files.

Reviewer #1: No

Reviewer #2: No

---

## [Author Response · Author response to Decision Letter 0]

2 Dec 2023

On behalf of the co-authors, we express our gratitude for the opportunity to revise our manuscript. We sincerely appreciate your insightful comments and suggestions on our manuscript titled "Exploring antecedents and outcomes of burnout among emergency department staff using the Job Demands-Resources model: A scoping review protocol" (No.: PONE-D-23-25522 ). We have thoroughly reviewed and diligently revised our manuscript following the reviewers' comments. The reviewer's comments are below in italicized font, and specific concerns have been numbered. Notably, our responses are in normal font, while additions to the manuscript are highlighted in green text. Moreover, all changes/additions to the latest manuscript use Microsoft Word's Track changes method. Below is a point-by-point response to the Editor and Reviewers' comments and concerns.

Responses to the comments of the Editor:

Comment 1: Please ensure that your manuscript meets PLOS ONE's style requirements, including those for file naming. 

Response: Thank you for your reminder. We have thoroughly reviewed and revised the manuscript style and file names in accordance with the journal's style requirements.

Comment 2: We noticed you have some minor occurrence of overlapping text with the following previous publication(s), which needs to be addressed:- https://doi.org/10.1111/1742-6723.12135. In your revision ensure you cite all your sources (including your own works), and quote or rephrase any duplicated text outside the methods section. 

Response: In response to these concerns, we have carefully rephrased the duplicated text and cited all sources involved(Lines 58~64).

Comment 3: PLOS requires an ORCID iD for the corresponding author in Editorial Manager on papers submitted after December 6, 2016. Please ensure that you have an ORCID iD and that it is validated in Editorial Manager. 

Response: In compliance with this requirement, I have updated my information and linked my ORCID iD with the Editorial Manager system for our submission.

Comment 4: Please review your reference list to ensure that it is complete and correct. 

Response: Thank you for your reminder. We have examined each citation in our manuscript to ensure the reference list is complete and correct.

Responses to additional Editor comments:

Comment: Information regarding burnout could be found in articile Who Cares What the Doctor Feels: The Responsibility of Health Politics for Burnout in the Pandemic. Kindly review and evaluate the requested work to determine whether they are relevant and should be cited. Once you have completed the revisions, please submit the revised version of your manuscript through our online submission system. Include both a clean version and a tracked changes version to aid our review process.

Response: Thank you for your valuable feedback and for providing the reference. We have carefully reviewed the article to assess its relevance to our work and cited this article in our revised manuscript, ensuring that it supports our findings(Lines 55~57).

As instructed, we are submitting a clean version and a tracked changes version of our revised manuscript through the online submission system.

Responses to the comments of Reviewer #1:

Comment: The manuscript appears to be well structured, overall. A suggested minor edit is to offer a concise overview of the anticipated benefits and implications of the scoping review in the Introduction section. Explain how the review's outcomes could affect healthcare organizations or practitioners. This background can assist readers grasp the significance of your findings.

Response: Thank you for your suggestion. We've revised the Introduction to include a concise overview of our scoping review's anticipated benefits and implications (Lines 92~98).

Responses to the comments of Reviewer #2:

Comment 1: Introduction and Rationale: The Introduction clearly states the high prevalence of burnout among ED staff due to chronic job stress. However, it would benefit from more specific data or statistics to emphasize the magnitude of the problem. Also, it would be useful to briefly mention why the Job Demands-Resources (JD-R) model is chosen for this study.

Response: Thank you for your insightful feedback. We have supplemented the Introduction with additional specific data and statistics to highlight the prevalence of burnout among ED staff (Lines 49~55).

We've also briefly explained why the Job Demands-Resources (JD-R) model is particularly relevant and suitable for this study (Lines 68~72).

Comment 2: Methodology: The methodology section is thorough, explaining the use of the Joanna Briggs Institute methodology and the PRISMA-ScR checklist. However, it would be beneficial to include more detailed information on the criteria for selecting studies (e.g., year range, type of studies included).

Response: Thank you for your kind reminder. As suggested, we have updated the methodology section to include more detailed information on our study selection criteria. Specifically, we have clarified that the included study types are original articles employing quantitative and qualitative research methods. We have also defined the year range of the studies from inception to August 19, 2023(Lines 134~136).

---

## [Editor Report · Decision Letter 1]

21 Feb 2024

Exploring Antecedents and Outcomes of Burnout among Emergency Department Staff Using the Job Demands-Resources Model: A Scoping Review Protocol

PONE-D-23-25522R1

Dear dr 

We’re pleased to inform you that your manuscript has been judged scientifically suitable for publication and will be formally accepted for publication once it meets all outstanding technical requirements.

Kind regards,

PLOS ONE

Additional Editor Comments (optional):

Dear authors,

I am pleased to inform you that, following revision, your manuscript titled "Exploring Antecedents and Outcomes of Burnout among Emergency Department Staff Using the Job Demands-Resources Model: A Scoping Review Protocol" has been accepted for publication in PLOS ONE.
---

## [Editor Report · Acceptance letter]

5 Mar 2024

PONE-D-23-25522R1 

PLOS ONE

Dear Dr. Anita, 

I'm pleased to inform you that your manuscript has been deemed suitable for publication in PLOS ONE. Congratulations! Your manuscript is now being handed over to our production team.

Kind regards, 

on behalf of

Dr. Jasna Karacic Zanetti 

Academic Editor

PLOS ONE